# Air Pollution in Poland: A 2022 Narrative Review with Focus on Respiratory Diseases

**DOI:** 10.3390/ijerph19020895

**Published:** 2022-01-14

**Authors:** Wojciech Nazar, Marek Niedoszytko

**Affiliations:** 1Faculty of Medicine, Medical University of Gdańsk, Marii Skłodowskiej-Curie 3a, 80-210 Gdańsk, Poland; 2Department of Allergology, Medical University of Gdańsk, Smoluchowskiego 17, 80-214 Gdańsk, Poland; mnied@gumed.edu.pl

**Keywords:** ambient air pollution, respiratory diseases, asthma, atopy, spirometry, COVID-19, Poland

## Abstract

According to the World Bank Group, 36 of the 50 most polluted cities in the European Union are in Poland. Thus, ambient air pollution and its detrimental health effects are a matter of immense importance in Poland. This narrative review aims to analyse current findings on air pollution and health in Poland, with a focus on respiratory diseases, including COVID-19, as well as the Poles’ awareness of air pollution. PubMed, Scopus and Google Scholar databases were searched. In total, results from 71 research papers were summarized qualitatively. In Poland, increased air pollution levels are linked to increased general and respiratory disease mortality rates, higher prevalence of respiratory diseases, including asthma, lung cancer and COVID-19 infections, reduced forced expiratory volume in one second (FEV1) and forced vital capacity (FVC). The proximity of high traffic areas exacerbates respiratory health problems. People living in more polluted regions (south of Poland) and in the winter season have a higher level of air pollution awareness. There is an urgent need to reduce air pollution levels and increase public awareness of this threat. A larger number of multi-city studies are needed in Poland to consistently track the burden of diseases attributable to air pollution.

## 1. Introduction

Ambient air pollution is one of the most significant environmental issues affecting health and wellbeing. It is linked to increased mortality from cardiovascular and respiratory diseases, including lung cancer and chronic obstructive pulmonary disease (COPD), shortening life expectancy [1]. It is estimated that in Europe in 2019, about 307,000 premature deaths were caused by chronic exposure to fine particulate matter with a diameter of 10 micrometres or smaller (PM_10_) and to coarse particulate matter with a diameter of 2.5 micrometres or smaller (PM_2.5_) [2]. In addition, about 40,000 premature deaths were attributed to chronic nitrogen dioxide (NO_2_) exposure [2].

According to the World Bank Group, 36 of the 50 most polluted cities in the European Union are in Poland [3]. Moreover, Polish cities are among the urban areas with the highest estimated PM mortality burden [4]. Despite a consistent decrease in annual outdoor PM concentrations in the most polluted Polish cities in the recent decade, current values still do not meet the World Health Organisation (WHO) guidelines [5,6].

In 2021, the WHO reviewed their 2005 guidelines and reduced the maximum recommended mean annual PM_10_ concentrations to 15 μg/m^3^, PM_2.5_ to 5 μg/m^3^ and NO_2_ to 10 μg/m^3^ [7]. These values were greatly exceeded in Poland between 2016 and 2019, especially in the southern part of Poland and during the winter season [5]. In addition, even if the less restrictive interim 3 target values are considered, 15 μg/m^3^ (PM_2.5_), 30 μg/m^3^ (PM_10_), and 20 μg/m^3^ (NO_2_), the thresholds are still exceeded in the majority of polish cities [5,8]. Moreover, in 2021 the WHO recommended that the 24-h mean daily concentrations should not exceed 15 μg/m^3^ (PM_2.5_), 45 μg/m^3^ (PM_10_), and 25 μg/m^3^ (NO_2_), respectively [8]. 

As the levels of indoor air pollution in Poland are elevated, studies analysing the burden of diseases attributable to indoor air pollution will also be evaluated [9].

Thus, both ambient and indoor air pollution and its detrimental health effects are a matter of immense importance in Poland. Despite this, the growing volume of studies on Poles’ health and air quality has not been summarised yet. This paper aims to review current findings on the subject, focusing on respiratory diseases, including lung cancer and Coronavirus disease 2019 (COVID-19) infections. Moreover, this study aims to review Poles’ awareness of air pollution and its effects on health. 

## 2. Methods and Materials

This review was performed according to the guidelines regarding the writing of an unsystematic narrative review [10,11]. PubMed, Scopus and Google Scholar databases were searched for relevant articles. 

In December 2021, the authors conducted two separate database searches. Several combinations of the following keywords were used: 

1. describing air pollution: ‘air pollution’, ‘indoor air pollution’, ‘particulate matter’;

2. describing population: ‘children’, ‘adults’, ‘Poland’, ‘polish cities’;

3. describing health problems and health outcomes: ‘asthma’, ‘chronic obstructive pulmonary disease’, ‘respiratory diseases’, ‘COVID-19’, ‘atopy’, ‘allergy’, ‘traffic’, ‘hospitality rates’, ‘mortality rates’, ‘spirometry’, ‘awareness’.

No filters were used. The initial search returned 48,001 articles (Figure 1, layout was based on the graph available in the official PRIMSA 2020 guidelines) [12]. After the initial abstract screening, 200 articles were selected for full-text analysis. 

Full texts that were not available in English or Polish, not available online, not including the polish population, case reports, case series, abstracts as well as editorials were excluded from the evidence synthesis. The authors focused on the most recent articles regarding air pollution in Poland (published after the year 2010). However, if the article was considered an important contribution to the field then it was also included in the analysis. Therefore, there were no strict inclusion criteria.

In total, 71 studies were included in this narrative review. The articles were analysed, described and summarized qualitatively.

First of all, however, large multi-centre studies will be briefly discussed to provide a broader international perspective on air pollution and diseases attributable to this threat.

## 3. International Perspective on Air Pollution

According to the largest international studies, increases in outdoor air pollution levels are linked to increases in mortality rates [13,14,15,16,17,18,19]. Moreover, increased respiratory mortality rates are also associated with household (indoor) air pollution [20,21]. Increased numbers of total hospital admissions, as well as hospital admissions due to respiratory tract diseases, are also observed [22,23,24,25].

In children, large long-term follow-up studies link outdoor air pollution to increased incidence of asthma and wheezing [26,27,28]. Exposure to indoor air pollution seems to be responsible for reduced lung development [21]. In addition to that, it was found that prenatal exposure to air pollution influences vital health parameters in newborns [29,30].

In adults, high long-term exposure to ambient air pollution is also associated with a greater area of emphysema in computer tomography scans as well as lower FEV1 volumes during spirometry tests [31]. A similar effect was observed by Elbarbary et al., who reported that increases in PM_2.5_, PM_10_ and NO_2_ reduced FEV1 and FVC values [32]. 

Among adults, exposure to outdoor air pollution was also identified as a risk factor for COPD [33]. Moreover, ambient air pollution was also responsible for asthma exacerbations [34]. Increased prevalence of respiratory diseases, asthma and COPD were also associated with household (indoor) air pollution [20,21]. Increased levels of air pollutants were also linked to exacerbation of allergic rhinitis [35]. Recently, the impact of air pollution on asthma and rhinitis has been investigated in the POLLAR study [36].

Higher levels of outdoor as well as indoor air pollution levels were also associated with an increase in lung cancer incidence [20,37]. Moreover, living in the proximity of high traffic areas was identified as an important risk factor for lung cancer [37].

Similar to the results obtained in Poland, the spread and mortality of COVID-19 seemed to be aggravated by air pollution in international studies [38,39].

Overall, the findings of Polish researchers seem to be consistent with the observations and conclusions drawn in large, international studies.

## 4. General and Respiratory Disease Hospitalization and Mortality Rates

Jastrzębie-Zdrój and Rybnik (Figure 2 and Figure 3) have one of the highest estimated age-standardised mortality rates per 100,000 population in Europe, preventable if the 2005 WHO guidelines for annual PM_2.5_ concentrations were met [4]. In the Silesian Voivodeship as a whole, it is estimated that the reduction of PM_10_ short-term exposure by 5 μg/m^3^ would result in a lower number of annual deaths due to non-external causes (2.60–2.75 per 100,000 inhabitants) [40]. 

For Kraków, in 2013, it was calculated that PM_10_ contributed to about 30 additional deaths per 100,000 inhabitants [41] (Figure 4). At the same time, 45 additional deaths per 100,000 were attributable to PM_2.5_, while increased NO_2_ concentration resulted in 9 additional deaths per 100,000. Thus, the estimated overall mortality due to PM_2.5_ was higher than that due to PM_10_ [41].

Holnicki et al. showed that in Warsaw, the capital city of Poland, local emissions cause approximately 1600 attributable deaths and 29,000 disability-adjusted life-years (DALYs) per year and that about 80% of this health burden was due to exposure to PM_2.5_ [42]. Maciejewska report that in Warsaw, a short-term exposure to increased PM_2.5_ and PM_10_ concentrations by 10 µg/m^3^ results in an increase of relative risk (RR) of death by 0.7% and 0.3%, respectively [43].

A detrimental impact of ambient air pollution was also observed in Łomża and Suwałki, which are located in the north-eastern part of Poland (Figure 1). In the years 2008–2017, an impact of both PM_10_ and PM_2.5_ on general mortality was reported [44]. In nearby Białystok it was shown that an increase in sulfur dioxide (SO_2_) concentration by 1 µg/m^3^ was related to an increase in the number of daily deaths (RR 1.07; 95% confidence interval (CI) 1.02–1.12). Interestingly, results for PM_2.5_ and PM_10_ were not significant [45], even though the change in PM_2.5_ concentration was shown to have the greatest strength of association with daily mortality rates [16,17]. However, a rise of PM_2.5_ concentration by 10 µg/m^3^ was found to be a risk factor for increased cardiovascular deaths (RR 1.07; 95% CI 1.02–1.12) [45].

Slama et al. analyzed the effects of air pollution on hospitalization rates due to respiratory diseases in the years 2014–2017 using data for Gdańsk, Białystok, Warsaw, Bielsko-Biała and Kraków, cities accounting for more than 20 million hospitalizations and about 10% of the total population of Poland [46,47]. For Gdańsk, Warsaw and Białystok, the estimated percentage increase in respiratory disease hospitalizations was equal to about 4% per 10 μg/m^3^ increase in PM_2.5_ concentrations [46,47]. For Kraków and Bielsko-Biała, a 10 μg/m^3^ rise in SO_2_ concentration correlated with a 5% rise in hospitalizations. An increased PM_2.5_ concentration was also linked to an increase in mortality rates due to respiratory diseases in Białystok [44]. 

Kowalska et al. reported that in years 2016–2017, during days with high PM_2.5_ and PM_10_ concentrations, a significantly higher number of respiratory disease hospitalizations and outpatient visits in hospitals of the Silesian Agglomeration (Bielsko-Biała and Katowice) was observed [48].

In Warsaw, it was estimated that about 230 hospitalizations p.a. due to respiratory problems in children were attributable to air pollution [49]. Moreover, a statistically significant estimated increase in general post-neonatal mortality and in post-natal mortality due to respiratory disease attributable to high air pollution levels was observed [49]. 

Overall, it seems that an increase in air pollution is linked to an increase in general and respiratory disease mortality rates across the whole country as well as hospitalization rates due to respiratory disease. PM_2.5_ concentration seems to be the most important factor affecting the general mortality rates. 

## 5. Children

### 5.1. Spirometry Outcomes

It was found that compared to a group of children aged 9–15 living in Gdynia, a city located on the Baltic coast, adolescents living in Zabrze had significantly worse spirometry outcomes [50]. The difference was observed for forced vital capacity (FVC) and forced expiratory volume in one second of expiration (FEV1) [50]. 

A similar effect was noticed by Jędrychowski et al., who reported that 9-year-old boys living in highly polluted districts of Kraków had lower basic FVC and FEV1 as well as a smaller increase in these over two years of growth compared to boys living in less polluted areas [51]. The same tendency was observed among girls, but only the differences in FVCs were statistically significant [51]. In another study, Jędrychowski also found that both FVC and FEV1 could be impaired in pre-adolescent children (9-years-old) with high exposure to poor indoor air quality in the postnatal period, while high prenatal exposure to ambient air pollution was measured by changes in PM_2.5_ concentrations, resulting in lower FVC and FEV1 levels measured for 5-year-olds [52].

In another prospective study which analysed the relationship between prenatal exposure to polycyclic aromatic hydrocarbons (PAH) and impaired lung function of 5–8-year-old children living in Kraków, a significant relationship was found for FEV1. Additionally, both indoor and outdoor exposure to PAH was found to exacerbate the effect of prenatal exposure [53].

Another prospective study showed that non-asthmatic children aged 4–9 years living in Kraków, who were prenatally exposed to higher levels of PM_2.5_ as PAH, had significantly lower initial values of FVC and FEV1 compared to the control group [54] and that the rate of increase in both parameters remained similar. Therefore, it seems that the burden of the initial effect of prenatal exposure remains uncompensated over time [54].

Recently, it was reported that antihistamine medications could alleviate the negative effects of prenatal exposure to PAH on lung function in children, as in the group of antihistamine users no statistically significant relationship was found between prenatal PAH exposure and lung function impairment [55]. This suggests that PAH exposure may cause allergic inflammation within the lungs [55].

Siniarska et al. reported that in adolescents aged 13–16 years, air pollution was related to lower minute ventilation (MV), longer apnoea (Ap) and higher inspiratory reserve volume (IRV) in children living in the more polluted areas of Warsaw [56]. However, the differences in FEV1 and FVC were not statistically significant [56]. 

On the other hand, Zejda et al. found no statistically significant relationship between FVC and FEV1 and air pollution in boys aged 7–9 years [57]. However, the authors highlight numerous methodological limitations of the study which might have influenced the obtained results [57]. 

To sum up, studies carried out in Poland report that children living in the areas with greater levels of air pollution have lower FVC, FEV1 and MV, longer Ap and higher IRV values compared to the control groups (Figure 5). Moreover, children who were pre- and postnatally exposed to high PM_2.5_ and/or PAH levels have impaired lung function, too, and the initial damage to the lung tissue does not diminish over time. Finally, some results suggest that antihistamine treatment may alleviate lung function impairment in children exposed to high PAH levels.

### 5.2. Respiratory and Allergic Symptoms

In 1996, Zejda et al. reported that in children aged 7–9 years, the frequencies of wheezing, attacks of asthmatic dyspnoea as well as the diagnosis of asthma were more prevalent in the city of Chorzów in comparison with controls from less polluted areas [58]. Moreover, a positive relationship between the prevalence of bronchitis and bronchiolitis and the combined effect of a rise in PM and SO_2_ concentrations was observed in Chorzów [59]. For children living in Zabrze, an increased prevalence of seasonal rhinorrhoea and cough episodes was observed [50]. Brożek et al. observed that an improvement of ambient air quality, also in Chorzów, correlated with a decreasing prevalence of cough [60].

Children living in highly polluted areas of Kraków experienced wheezing, allergic rhinitis, dyspnoea attacks and chronic cough more frequently than control groups [61,62]. Both outdoor and indoor air pollution also increased the risk of hay fever [63]. Moreover, the overall risk of allergy in children was increased [62]. 

In addition, a relationship between prenatal exposure to ambient PAH and an increased prevalence of cough independent of respiratory tract infections, wheezing, sore throat and ear infections were observed [64]. Moreover, the duration of runny nose, cough and other difficulties in breathing was prolonged in newborns exposed prenatally to the worse quality air [64]. More recently, it was found that increased intrauterine exposure to high PM_2.5_ concentrations was related to an increased incidence of recurrent broncho-pulmonary infections as well as the severity of respiratory illness in early childhood [65,66]. Additionally, Dyląg et al. observed that a rise in PM_2.5_, PM_10_ and nitrogen oxides (NO_x_) concentrations was positively associated with the incidence of viral croup in Kraków [67]. However, another study found no direct relationship between prenatal exposure to PM_2.5_ and the onset of early wheezing in infants [66].

In a study carried out in Warsaw, it was estimated that about 5200 additional asthma symptoms in children are annually attributable to air pollution [49]. Similar to studies carried out in the southern part of Poland, greater risks of cough, bronchitis, lower respiratory symptoms as well as a higher incidence of asthma symptoms in asthmatic children were associated with higher air pollution levels [49].

According to the most recent nationwide studies, increased frequency of non-allergic rhinitis in children aged 6–14 years old was associated with the use of indoor heating systems utilizing gas or solid fuels, suggesting a relationship between low indoor air quality and an increased risk of non-allergic rhinitis [68]. Moreover, indoor air quality was also found to be a risk factor for asthma [69].

In children aged 3–12 years old, the prevalence of upper respiratory tract symptoms such as a runny nose, sneezing and cough was positively associated with high 12-week mean PM_2.5_ and PM_10_ concentrations [70]. The difference between the frequency of symptoms between the control and the most polluted regions was about 10% [70]. Moreover, Wrotek et al. reported that paediatric hospitalizations due to respiratory syncytial virus infections in Poland were significantly associated with a rise in either PM_2.5_, PM_10_ or NO_2_ concentrations [71].

Overall, the studies reviewed suggest that both indoor and outdoor air pollution have an impact on the prevalence and duration of respiratory symptoms (attacks of wheezing, dyspnoea, bronchitis, chronic cough) in infants and children living in Poland. Increased risks for asthma, allergic rhinitis and respiratory tract infections were also observed. The trends have remained mostly unchanged over the past 20 years.

### 5.3. Traffic Zone Proximity

Interestingly, exacerbation of respiratory health problems seems to be higher in children living close to high traffic areas. In Krakow, children aged 7 and 16-years-old living within 200 m of major roadways complained more often of sneezing, runny or blocked nose accompanied by itchy-watery eyes and hay fever in comparison to subjects living 200 to 500 m from a major roadway. The lowest rate of nasal symptoms was observed for residents living a large distance from major roads (>500 m) [72]. 

In a study performed in Bytom, a statistically significant relationship was found between the prevalence of asthma and residential proximity to traffic [73]. Similar relationships were recorded for allergic rhinitis and rhinitis symptoms, but the associations were not statistically significant [73]. Adolescents aged 13–15 years old living in Chorzów also reported a statistically significant higher prevalence of asthma and allergic rhinitis [74]. Increased occurrence of wheezing and dyspnoea attacks in patients living in the vicinity of the main road were also noticed [74]. However, the lung function indices as well as skin prick test results did not differ between groups [74].

To sum up, living close to major roadways exacerbates the respiratory health problems in children. The prevalence of allergic rhinitis and its symptoms as well as the prevalence of asthma are also higher.

## 6. Adults

### 6.1. Spirometry Outcomes

In a study covering Wrocław and Kraków, assessing how ambient exposure in childhood and adolescence affects lung function in adulthood, it was observed that FEV1 and FVC were reduced in people living in large cities compared to villages and smaller towns [75]. Reduced FVC and FEV1 were also associated with exposure to high PM_10_, PM_2.5_, NO_2_ and CO concentrations [75]. Another study that compared people living in rural and urban areas showed that FEV1 decreased about 1.7% for each 10 μg/m^3^; increase in PM_10_ [76]. Lubiński et al. reported that in a group of young, healthy, non-smoking men living in more polluted areas, airflow limitation both in the central bronchi and in the small bronchi was observed more frequently in comparison to the control group [77]. A negative relationship between NO_2_, SO_2_, and PM_10_ and several lung function parameters (FVC, FEV1) was observed [77]. Dąbrowiecki et al. reported that among people living in cities and close to the main roads, an increased number of obstruction cases, lower FEV1 as well as decreased FEV1/FVC were observed [78]. 

Kocot et al. reported that the higher the PM_2.5_ and/or SO_2_ concentration during exercise, the greater the decrease in post-exercise FEV1/FVC [79], although another study from the same author did not describe significant associations between exposure to air pollution and FVC and FEV1 [80] or pulse oximetry measurements [79,80]. As both studies concerned young, healthy males, differences may still exist in other age and gender groups in Poland. Kocot et al. also observed a statistically significant rise in fractioned exhaled nitric oxide levels in participants exposed to high air pollution in comparison to the control group [80,81]. All the discussed studies were carried out in Katowice [79,80,81].

In the analysis comparing inhabitants of Warsaw and rural areas, Badyda et al. reported significantly lower values of FEV1 and FEV1/FVC in Warsaw [82]. The same study identified living in the vicinity of high traffic areas as a risk factor for lower overall spirometry outcomes [82]. On the contrary, physical activity was responsible for a positive effect on pulmonary function (higher FEV1) and partially reduced the negative health effects of traffic-related emissions [82].

To sum up, similarly to the spirometry parameter variations observed in children living in highly polluted areas, also studies investigating adults living in Poland report lower FEV1, FVC and FEV1/FVC values associated with higher ambient air pollution (Figure 6). The strongest relationships were observed for PM_10_, PM_2.5_, NO_2_ and SO_2_. The proximity of high traffic areas was also a negative factor influencing lung function. The influence of air pollution on the lung function of young, healthy males during exercise remains unclear.

### 6.2. Bronchitis

Outpatient visits due to bronchitis in the Silesian Voivodeship were positively associated with the rise in NO_2_ concentrations measured using interquartile (IQR) range (RR 1.43; 95% CI 1.31–1.57) [83]. NO_x_ (RR 1.36; 95% CI 1.26–1.47) and PM_2.5_ (RR 1.07; 95% CI 1.02–1.12) were also significantly associated with the number of outpatient visits [83]. The RR values were even higher for hospitalization: RR 1.67; RR 1.51; RR 1.19, respectively [83].

Also in Silesia, a study covering the years 2016–2017 reported higher ozone concentrations associated with an increased risk of hospitalizations due to acute bronchitis (RR 1.15; 95% CI 1.03–1.28) [84]. During a smog episode in the Silesian Voivodeship in January 2017, a positive relationship between PM_2.5_ concentrations and a number of outpatient visits due to exacerbation of bronchitis was also observed [85]. 

### 6.3. Asthma

In a nationwide PMSEAD study including over 16,000 subjects living in 11 regions of Poland, residential exposure to traffic-related air pollution in both children and adults was identified as a risk factor for asthma [86]. However, no significant relationship between ambient air pollution with SO_2_ and the prevalence of asthma was observed [86].

A study involving patients living in Kraków exposed to high PM_2.5_ concentrations reported a significantly lower total score in the Asthma Quality of Life Questionnaire [87]. The pollution had a negative effect on all domains of quality of life: prevalence of symptoms, activity limitations, emotional functioning and environmental stimuli [87]. Moreover, an increased number of asthma hospitalizations was associated with increased SO_2_ concentrations (RR 1.05; 95% CI 1.03–1.08) and PM_10_ (RR 1.02; 95% CI 1.01–1.03) [88].

In Katowice, in January 2017, an acute PM_2.5_ concentration increase was followed by more frequent outpatient visits due to asthma exacerbation [85]. In addition, in the 2016–2017 period, an increased concentration of NO_2_ measured by its IQR range (RR 1.23; 95% CI 1.14–1.33), NO_x_ (RR 1.18; 95% CI 1.11–1.26) and PM_2.5_ (RR 1.06; 95% CI 1.02–1.10) was related to a greater number of outpatient visits due to asthma [83]. The RR values for hospitalization were even higher: RR 1.37; RR 1.30; RR 1.07, respectively [83].

There is also evidence of the effects of indoor air pollution on the prevalence of asthma. In a study carried out in Warsaw, the use of a solid-fuel heating system (OR 1.36; 95% CI 1.05–1.75), as well as the use of cooking appliances supplied with municipal natural gas (OR 1.77; 95% CI 1.06–2.97) or gas storage tanks (OR 2.03; 95% CI 1.19–3.45), were correlated with respondents reporting asthma more frequently [69].

Overall, studies provide ample evidence of exacerbation of both asthma and bronchitis symptoms due to higher outdoor concentrations of air pollutants (NO_2_, NO_x_, PM_2.5_) resulting in a greater number of outpatient visits and hospitalizations and a decreased quality of life of people with asthma. Exposure to traffic-related air pollution as well as worse indoor air quality were also identified as risk factors for asthma.

### 6.4. Atopy

In Szczecin, a city in the north-eastern part of Poland, the presence of air pollutants such as ozone, PM_10_ and SO_2_ was significantly associated with higher pollen counts [89]. Therefore, air pollution may contribute to allergy exacerbation in atopic subjects. 

Czarnobilska et al. showed that 83% of atopic subjects and 75% of non-atopic patients had positive results of the basophil activation test in response to PM_2.5_ stimulation [90]. Moreover, in atopic patients, much greater basophil activation test scores were obtained for simultaneous birch allergen and PM_2.5_ provocation tests than the sum of scores for consecutive exposures to birch allergen and PM_2.5_ individually [90]. Thus, air pollution seems to sensitise atopic patients and have a synergistic effect when combined with allergen exposure, triggering a much stronger basophil response.

### 6.5. Rhinitis

In a nationwide study on the effects of indoor air quality on the development of rhinitis in the urban population in Poland, the increased prevalence of allergic rhinitis in subjects aged 20–44 years old was associated with the presence of a gas furnace used to heat the house (OR 1.19; 95% 1.05–1.34) as well as the use of a solid-fuel stove (OR 1.92; 95% CI 1.07–3.46) and/or bottled-gas stove (OR 1.66; 95% CI 1.28–2.16) [68]. Risk factors for prevalence of non-allergic rhinitis included the presence of gas stove heating (OR 1.27; 95% CI 1.14–1.41), a solid-fuel stove (OR 2.02; 95% CI 1.29–3.18) and/or a bottled-gas stove (OR 2.06; 95% CI 1.66–2.55) in the household [68]. Moreover, Tomaszewska et al. reported that allergic rhinitis was more prevalent in the urban than in the rural population of Poland [91]. This suggests that air pollution may contribute to the development of allergic rhinitis. On the other hand, no significant relationships were found between the duration of birch pollen season, which may be exaggerated by high air pollution [89], and the prevalence of declared allergic rhinitis during the season [92].

### 6.6. Lung Cancer

In a study covering a group of males in the Silesian Voivodeship, a relationship between an increased PM_10_ concentration and an increased lung cancer morbidity was observed [93]. A more recent study performed by Badyda et al. found that PM_2.5_ may also contribute to lung cancer mortality rates in Poland [94]. Moreover, they estimated that in the years 2006–2011, in eleven large Polish cities, about 20% to 40% of lung cancer cases seemed to be attributable to chronic PM_2.5_ exposure [94]. Chudzik et al. estimate that in the years 2009–2017, on average, each Pole “smoked” an equivalent of about 10 cigarettes a day [95]. Thus, every Pole over 60 years old will reach the “30 pack-years of smoking”, which is considered a threshold for being classified as a high-risk patient and is an indication for low-dose computer tomography screening [95,96]. Therefore, ambient air pollution may be considered an independent risk factor for lung cancer in Poland [95]. Due to an inadequate number of studies regarding air pollution and lung cancer morbidity and mortality in Poland, a need for further research was previously stipulated [97]. As of 2021, the data regarding this problem is scarce and future environmental and epidemiological investigations are highly desirable. 

### 6.7. Traffic Zone Proximity

In a study performed in Kraków, it was found that adult atopic residents of the city living closer to main roads had higher urine concentrations of 1-hydroxypirene (1-OHP than those living farther away from high traffic areas [90]. As the presence of 1-OHP in the urine of non-smoking subjects reflects exposure to benzopyrene, it was concluded that people living in high traffic areas were exposed to greater levels of this pollutant [90]. Moreover, as discussed earlier, the residential proximity of main roads worsens overall spirometry outcomes and is a risk factor for asthma [82,86].

## 7. COVID-19 Pandemic

### 7.1. National Lockdown in 2020 in Poland

During the national lockdown in April and May 2020, a reduction of aerosol concentration in the air column of approximately −23% and −18% as compared to 2018–2019 was observed [98]. Interestingly, the greatest contraction was noted for PM_2.5_ with a reduction of about −15% to −25% in April and −15% in May 2020. For PM_10_ the reductions ranged from about −10% to about −30% in this period [98]. A study in Lublin, a city located in the eastern part of Poland, reported a substantial decrease in traffic intensity during lockdown restrictions, resulting in a significant reduction in the concentration of traffic-generated particles [99].

### 7.2. COVID-19 Incidence and Mortality Rates

At the beginning of the pandemic in Poland, for the March–May 2020 period, statistically significant correlations between the number of COVID-19 cases per 100,000 people and the annual average concentration of PM_2.5_ (R^2^  =  0.367), PM_10_ (R^2^  =  0.415), SO_2_ (R^2^  =  0.489), and ozone (R^2^  =  0.537) were established [100]. COVID-19 deaths per 100,000 population were also associated with an annual average concentration of PM_2.5_ (R^2^  =  0.290), NO_2_ (R^2^  = 0.319) and ozone (R^2^  =  0.452) [100]. 

In a similar study conducted by Musiałek et al., a relatively strong correlation between NO_2_ concentration and COVID-19 age-adjusted mortality rate per 100,000 population (ρ = 0.60, *p* < 0.05) was observed [101]. For PM_2.5_ concentrations, a non-statistically significant correlation was reported (ρ = 0.46) [101].

Moreover, according to a numerical analysis of factors influencing the pace and intensity of the pandemic, air quality measured by PM_2.5_ concentration variations was the most important factor associated with COVID-19 caseload increase in Poland [102]. 

Interestingly, between 2013 and 2019 in Białystok, an exponential relationship between rising cumulative PM_2.5_ concentration and incidence of influenza-like illness was established [103]. Thus, it seems that a decrease in PM_2.5_ concentration would be beneficial to reduce the transmission of severe acute respiratory syndrome coronavirus-2 (SARS-CoV-2) and other respiratory tract infections [103].

To conclude, it seems that an increase in PM_2.5_, PM_10_, SO_2_, NO_2_ and ozone concentrations may enhance the spread and rate of the infectiousness of SARS-CoV-2. Moreover, high NO_2_ and PM_2.5_ concentrations seem to be connected with an increased COVID-19 mortality rate.

## 8. Social Awareness of Air Pollution in Poland

There is a need for increased awareness among both the general public and medical professionals of the burden of disease attributable to air pollution. In a previous study, we found that in general, the Poles living in the southern part of Poland search the internet more frequently for air pollution-related terms [5]. Moreover, the relative keyword search intensity increased during winter, when air pollution is generally higher [5]. Thus, people seem to respond behaviourally to elevated air pollution levels. We also found that the information-seeking behaviour correlates more strongly with the PM_2.5_ and PM_10_ concentration measurements performed in large cities and urban areas, compared to measurements carried out for the whole voivodeship. Therefore, it seems that people living in urban areas are more conscious of air pollution [5].

According to recent data, only about 25% of Polish physicians think their knowledge of the impact of air pollution on health is sufficient [104]. Only 3% of them inform their patients when air pollution exceeds permissible limits [104]. Moreover, among pulmonologists, only about 15% declare sufficient knowledge on air pollution [105]. Whilst pulmonologists declare more frequently that they have broad knowledge on diseases caused by air pollution compared to the patients, their actual knowledge is not greater. Thus, patients seem to have greater knowledge on the topic than professionals [105]. For both groups, the internet and radio/TV are among the most common sources of information, but for pulmonologists medical press remains the most important source [105].

To increase the awareness of air pollution, alerts during days with relatively high pollution levels can be provided. Adamkiewicz et al. suggest to implement national information and alert thresholds equal to 64 µg/m^3^ and 83 µg/m^3^ of the daily mean PM_10_ concentrations, respectively [106]. Theoretically, reduction of exposure to air pollution during these days would reduce the burden of hospital admissions attributable to air pollution by 75% and 50%, in regard to the best case scenario (no days with PM_10_ concentration exceeding 50 µg/m^3^).

Overall, Poles seem to be aware of the air pollution problem, with people living in more polluted areas and during the winter season manifesting intensified information-seeking behaviour in comparison to people living in less polluted areas (Figure 7). However, studies on this topic are scarce. The need for professional education on air pollution among medical professionals in Poland appears to be quite acute. Moreover, to avoid the need for hospitalizations attributable to air pollution, an implementation of scientifically derived national information and alert thresholds is needed.

## 9. Limitations and Future Perspectives

Due to various methodological differences across the analysed studies, only a qualitative evaluation without a quantitative summary is possible. The differences between the studies include the adoption of a large number of different outcome points (mortality, hospitalisation rates), metrics (RR, OR, spirometry tests) and threshold values (IQR, the 10 units difference). Moreover, different inclusion and exclusion criteria were used to define high and low polluted areas. In addition, researchers adopted a varied selection of pollutants (PM_10_, PM_2.5_, SO_2_, NO_x_ or PAH) as the reference component of air pollution level. Moreover, the adopted approach shows that air pollution impacts a large number of human physiology parameters and may play a role in various epidemiological mathematical models.

Secondly, most of the reviewed studies are cross-sectional investigations, usually without any type of follow up. Numerous studies include only two or three groups that were compared to draw conclusions. Additionally, the majority of them were conducted in the southern and south-western parts of Poland. Thus, a large spatial discrepancy exists and the northern part of Poland remains underexplored. Nevertheless, in cities like Białystok (north-east) or Gdańsk (north) a detrimental effect of air pollution on health was also investigated and reported, which further supports the need for more research in those areas.

A possible solution for the abovementioned imperfections would be multi-city multi-outcome studies. These would allow for reliable comparisons of urban and rural areas across the whole country and conclusions drawn from the investigation would be highly valuable. However, such an approach is rarely possible due to methodological difficulties, the effort and the cost of such projects.

Moreover, only a few studies investigated the awareness and air pollution-related information-seeking behaviour of Poles. Thus, there is very little evidence on how the population reacts to occasionally elevated air pollution levels.

In addition, it should be acknowledged that this study is a narrative review, not a systematic review. Therefore, the selection of studies is prone to bias and is based on authors’ judgments. The authors aimed to reduce the selection bias and described their selection criteria in the Methods and Materials section. Moreover, the conclusions drawn in this review are susceptible to potential publication bias regarding cited literature [107]. To reduce this, many databases were searched for relevant articles.

On a more positive side, the variety of existing studies allowed for the inclusion and comparison of research concerning infant, children, adolescent and adult subpopulations. Moreover, the combination of epidemiologic (indirect sampling) and in-person (direct sampling) sampling allowed for a more comprehensive analysis of the evidence on health problems caused by air pollution.

## 10. Conclusions

In Poland, increased air pollution levels are linked to increased general and respiratory disease mortality and hospitalization rates as well as a higher prevalence of respiratory symptoms (wheezing, dyspnoea attacks, chronic cough) and respiratory diseases, including bronchitis, asthma, rhinitis, lung cancer and COVID-19 infections. Reduced FEV1 and FVC in spirometry tests are also observed. Infants, children, adolescents as well as adults are affected by worse air quality. Living in the proximity of high traffic areas exacerbates respiratory health problems. 

There is ample evidence that the course of asthma and bronchitis is exacerbated in areas with high ambient air pollution, while the indicators of subjective quality of life in asthmatic patients are lower. In atopic subjects, air pollution seems to sensitise the body and trigger an excessive response when combined with the usual allergen. There is also some evidence that higher concentrations of PM_2.5_, PM_10_, SO_2_, NO_2_ and ozone may facilitate the spread and infectiousness of SARS-CoV-2. 

Concerning the public awareness of air pollution, people living in more polluted regions (south of Poland) and during the winter season are more aware of air pollution levels. 

Most commonly studied air pollutants include PM_10_, PM_2.5_, NO_x_, NO_2_, SO_2_, ozone and PAH. Across the reviewed studies, we found large methodological differences, as various inclusion and exclusion criteria, metrics and outcomes were adopted. This allows for qualitative summary analysis only. Moreover, most of the research was performed in the southern part of Poland. Thus, there is a need for large, multi-city multi-outcome studies to draw reliable comparisons.

From an international perspective, the findings of Polish researchers are consistent with the observations and conclusions drawn in large studies elsewhere.

Overall, due to its various detrimental effects on health, the authors recognise an urgent need to reduce the air pollution level. Therefore, in order to minimize the air pollution levels, a more advanced national pro-ecological policy must be implemented and a shift from coal towards low-carbon energy sources must be a endorsed. This should be coupled with intensive promotion of social awareness regarding this issue. Moreover, thorough multi-factorial multi-city investigations are needed to analyse the burden of diseases attributable to air pollution comprehensively. 

## Figures and Tables

**Figure 1 ijerph-19-00895-f001:**
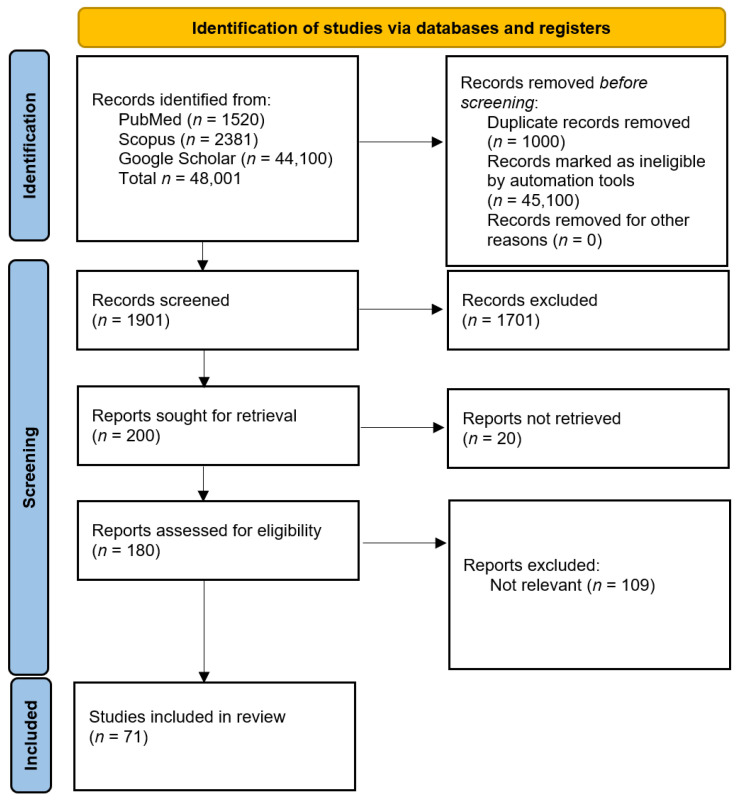
Studies selection.

**Figure 2 ijerph-19-00895-f002:**
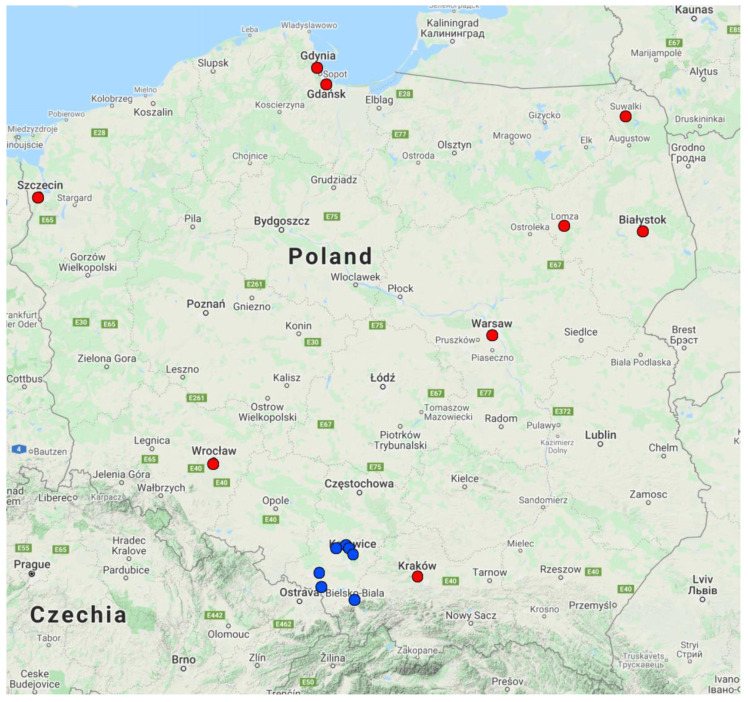
Location of cities mentioned in this review. Blue dots represent cities located in the Silesian Voivodeship. Red dots represent all other cities.

**Figure 3 ijerph-19-00895-f003:**
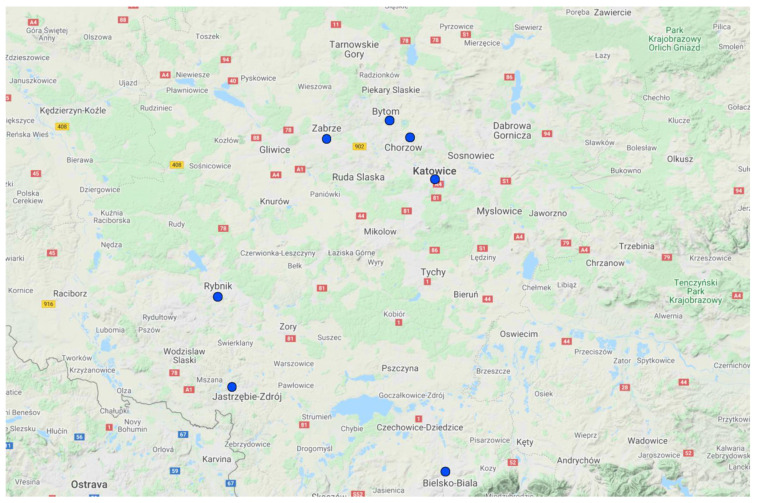
Cities located in the Silesian Voivodeship.

**Figure 4 ijerph-19-00895-f004:**
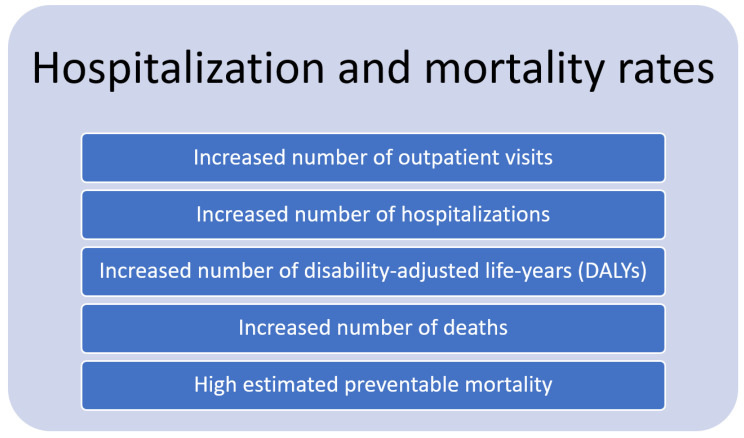
Effect of air pollution on hospitalization and mortality rates.

**Figure 5 ijerph-19-00895-f005:**
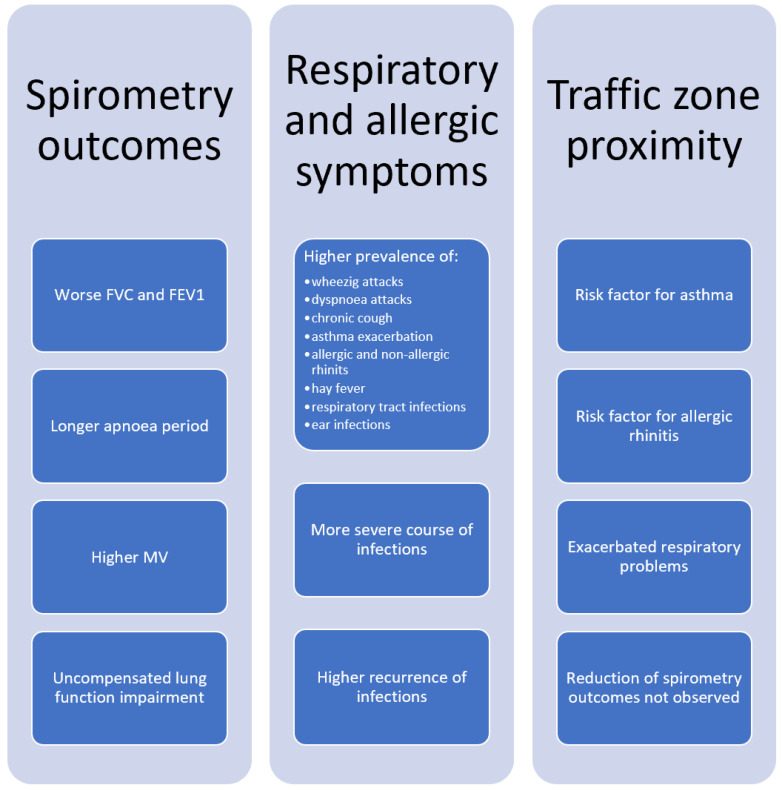
Effect of air pollution on children’s health.

**Figure 6 ijerph-19-00895-f006:**
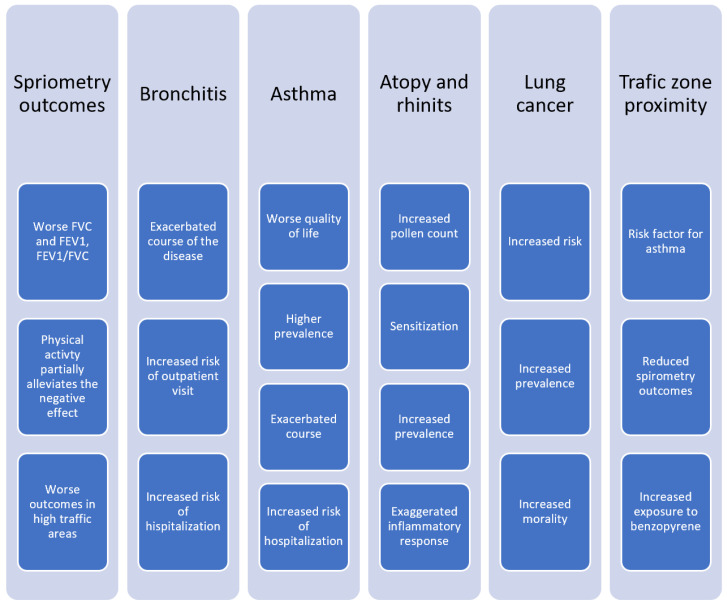
Effect of air pollution on adults’ health.

**Figure 7 ijerph-19-00895-f007:**
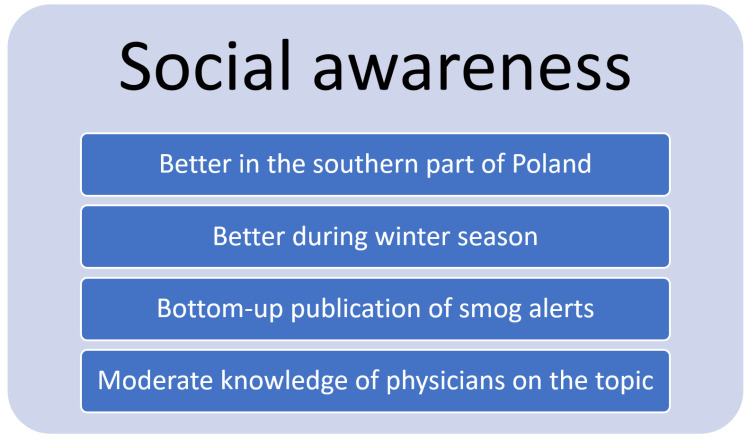
Characteristics of social awareness of air pollution in Poland.

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
