# Peer review of "Air Pollution in Poland: A 2022 Narrative Review with Focus on Respiratory Diseases"

_ijerph, 2022, doi:10.3390/ijerph19020895_

Round 1

Reviewer 1 Report

I appreciate to have an opportunity to review this paper. The paper is based on significance of the research question regarding air pollution and health issues in Poland, which is known to be highly polluted regions in Europe. This research aims to review recent findings for the associations between air pollution and health outcomes and social awareness on air pollution in Poland. While a significant number of papers were reviewed in this article, important elements of study methods are missing in the current manuscript, which negatively affects transparency of this research. While this review looks like a narrative review, more organized structures of presentations and sections can help summarize research findings. Below is suggestions in detail.

  1. Some important elements of review should be mentioned in abstract. It is unclear if this review is systematic review, narrative review, topical review or any other, which should be clarified. How literature was searched and reviewed should be also mentioned. It is important if the findings were summarized qualitative or quantitatively. The number of papers reviewed in this research should be added. The last sentence (“a large number of multi-city studies…” is not reasonably supported by what is summarized in abstract.
  2. Page 1 line 28. Spell out COPD before using the abbreviation.
  3. Page 1 line 30. I suggest that PM10 and PM2.5 is more correctly mentioned here: particulate matter with a diameter of 10 micrometers or smaller (PM10).
  4. Awareness of air pollution and health was also reviewed in this manuscript. Such assessment may need to be mentioned as an aim of this research in the Introduction.
  5. This study does not have a method section, which is a critical issue for transparency and reproducibility of the research. Some important elements should be elaborated in this section: search database, search methods, search terms (keywords), search period, range of target outcomes and exposure factors, etc. The study design should be clarified as well. As mentioned before, it is not clear if this is a systematic review, narrative review, rapid review, or something other type of reviews. Important factors should be specified in accordance with the type of review.
  6. It will be helpful to have a table showing each of the reviewed papers and their study size, region, period, study design, and findings. For now, throughout the manuscript, the presentation of main findings of each study varies.
  7. Page 8 lines 257. Is this the only recent study for indoor air pollution in Poland? If indoor air pollution research is reviewed for asthma in adults, such review is expected in the section for children. Also, the reason why indoor air pollution is important (e.g., they are highly correlated with outdoor air pollution levels) should be mentioned in the manuscript.
  8. Authors provided figures showing the studied health outcomes in relation to air pollution in the reviewed studies in Poland. Do these figures cover all types of diseases that have been studied in relation to air pollution in previous reviews? What diseases have been mentioned in previous reviews and which of them have not been found in this manuscript? Explaining this will help identify diseases in Poland that need further research in the future.
  9. Is there a reason why section 5 (Traffic zone proximity) stands as a single section while the other sections are mainly categorized by health outcomes in certain ages? The papers in this section could be combined in section 3-4. Figure 6 can be misunderstood that only traffic zone proximity was reviewed as a measure of air pollution in this review.
  10. Section 8 (international perspective on air pollution) should be placed before the review of Polish studies. Also, for the sake of transparency, it should be elaborated why these international studies are compared to Polish studies. Review research other than systematic review is prone to bias from authors’ judgments, belief, or experts’ opinions.
  11. Page 12 lines 429. Do you mean that different inclusion/exclusion criteria were used across the included studies, or you had different inclusion/exclusion criteria for selecting studies to review? Having firm inclusion/exclusion criteria is crucial for review research. Authors may want to clarify better.
  12. Page 12 lines 436. It would be helpful if authors provide the number of cross-sectional studies out of the total included studies in this review.
  13. Authors should mention that this review research is prone to bias as it is not (presumably) a systematic review. Also, authors should mention potential publication bias for the studied outcomes and air pollution measures in Poland.
  14. Page 12 lines 433. It is not clear what exactly authors mean for ‘varied exploration establishes various successful frame works’. Authors may want to provide information on which exposure measures have been applied (e.g., daily monitoring data; satellite-based modeling data) in the reviewed studies. This would help understand data availability and potential misclassification for exposure measurement of air pollution in the reviewed Polish studies.

Author Response

I appreciate to have an opportunity to review this paper. The paper is based on significance of the research question regarding air pollution and health issues in Poland, which is known to be highly polluted regions in Europe. This research aims to review recent findings for the associations between air pollution and health outcomes and social awareness on air pollution in Poland. While a significant number of papers were reviewed in this article, important elements of study methods are missing in the current manuscript, which negatively affects the transparency of this research. While this review looks like a narrative review, more organized structures of presentations and sections can help summarize research findings. Below is suggestions in detail.

 ######################

Dear Reviewer, We appreciate the time and effort that you have dedicated to providing your valuable feedback on our manuscript. We are grateful for the insightful comments on our paper. We have been able to incorporate changes to reflect the very most of the suggestions provided (almost all). We have highlighted the changes within the manuscript.

We wrote a narrative review (not a systematic review).

  1. Some important elements of review should be mentioned in abstract. It is unclear if this review is systematic review, narrative review, topical review or any other, which should be clarified. How literature was searched and reviewed should be also mentioned. It is important if the findings were summarized qualitative or quantitatively. The number of papers reviewed in this research should be added. The last sentence (“a large number of multi-city studies…” is not reasonably supported by what is summarized in abstract.

######################

The abstract was adjusted according to the suggestions.

  1. Page 1 line 28. Spell out COPD before using the abbreviation.

######################

We spelled COPD out.

  1. Page 1 line 30. I suggest that PM10 and PM2.5 is more correctly mentioned here: particulate matter with a diameter of 10 micrometers or smaller (PM10).

######################

We changed the description

  1. Awareness of air pollution and health was also reviewed in this manuscript. Such assessment may need to be mentioned as an aim of this research in the Introduction.

######################

We added a sentence:

Moreover, this study aims to review Poles’ awareness of air pollution and its effects on health.

  1. This study does not have a method section, which is a critical issue for transparency and reproducibility of the research. Some important elements should be elaborated in this section: search database, search methods, search terms (keywords), search period, range of target outcomes and exposure factors, etc. The study design should be clarified as well. As mentioned before, it is not clear if this is a systematic review, narrative review, rapid review, or something other type of reviews. Important factors should be specified in accordance with the type of review.

######################

Our review is a narrative review. We added the Materials and Methods section to our manuscript to include there relevant information regarding the search strategy.

  1. It will be helpful to have a table showing each of the reviewed papers and their study size, region, period, study design, and findings. For now, throughout the manuscript, the presentation of main findings of each study varies.

######################

Dear Reviewer, we wrote a narrative review, not a systematic review. Therefore, in our manuscript we covered a very broad range of studies, with very large variety of different study designs. For example, some studies are based on satellite measurements, the other ones describe patients. Therefore, it is very hard to form a table that would encompass all possible study designs, as in different study designs different aspects are of high importance. For example, in studies involving patients, age, sex, comorbidities of participants are of high importance. Studies involving satellite measurements does not cover any of this aspect. Thus, a table would be full of “-“ marks, which in our opinion would not improve clarity of the study. An alternative to it could be many smaller tables. However, many articles have “one of its kind” study designs. It is hard to match them with the classical study designs like cross-sectional study or cohort study. Therefore, all in all only results of the studies can be compared. However, we discussed the results of the studies in our review. Therefore, the table would only repeat them. Moreover, as we do not write a systematic review, the table of all reviewed studies is not formally required.

Therefore, we think that the table creation is not necessary.

  1. Page 8 lines 257. Is this the only recent study for indoor air pollution in Poland? If indoor air pollution research is reviewed for asthma in adults, such review is expected in the section for children. Also, the reason why indoor air pollution is important (e.g., they are highly correlated with outdoor air pollution levels) should be mentioned in the manuscript.

######################

Dear Reviewer, for the section, where children are described we added a sentence:

“Moreover, indoor air quality was found to be a risk factor for asthma, too [43].”

To describe indoor air pollution, we added a sentence in introduction:

“Studies analysing the burden of diseases attributable to indoor air pollution will also be evaluated, as the levels of indoor air pollution in Poland are elevated, too .”

  1. Authors provided figures showing the studied health outcomes in relation to air pollution in the reviewed studies in Poland. Do these figures cover all types of diseases that have been studied in relation to air pollution in previous reviews? What diseases have been mentioned in previous reviews and which of them have not been found in this manuscript? Explaining this will help identify diseases in Poland that need further research in the future.

######################

According to our knowledge and database search, this is the first review considering mainly the air pollution and its health effects in Poland. Therefore, there are no previous reviews that can be compared to our review.

  1. Is there a reason why section 5 (Traffic zone proximity) stands as a single section while the other sections are mainly categorized by health outcomes in certain ages? The papers in this section could be combined in section 3-4. Figure 6 can be misunderstood that only traffic zone proximity was reviewed as a measure of air pollution in this review.

######################

Dear Reviewer, this section is now combined in sections 3-4, for children and adults separately. Figure 6 was redistributed to figures relevant to these sections.

  1. Section 8 (international perspective on air pollution) should be placed before the review of Polish studies. Also, for the sake of transparency, it should be elaborated why these international studies are compared to Polish studies. Review research other than systematic review is prone to bias from authors’ judgments, belief, or experts’ opinions.

######################

We changed the order of the sections. In methods we added a sentence:

“First of all, however, large international studies will be shortly discussed in order to provide a broader worldwide perspective on air pollution and diseases attributable to this threat.”

  1. Page 12 lines 429. Do you mean that different inclusion/exclusion criteria were used across the included studies, or you had different inclusion/exclusion criteria for selecting studies to review? Having firm inclusion/exclusion criteria is crucial for review research. Authors may want to clarify better.

######################

We discussed this issue in methods:

“Full texts that were not available in English or Polish, not available online, not including polish population, case reports, case series, abstracts as well as editorials were excluded from the evidence synthesis. The authors focused on the most recent articles regarding air pollution in Poland (published after the 2010 year). However, if the article was considered an important contribution to the field then it was also included in the analysis. Therefore, there were no strict inclusion criteria.”

In line 429 we want to endorse the variety of included studies, which results in a conclusion encompassing a broad perspective of ages (from infants to adults)

  1. Page 12 lines 436. It would be helpful if authors provide the number of cross-sectional studies out of the total included studies in this review.

######################

Dear Reviewer, many studies include something like “repeated cross-sectional” design. Therefore, it is hard to definitely state the number of cross-sectional studies, as it is prone to bias of our judgement, if a given study have a pure cross sectional design. This may be confusing to the reader.

  1. Authors should mention that this review research is prone to bias as it is not (presumably) a systematic review. Also, authors should mention potential publication bias for the studied outcomes and air pollution measures in Poland.

######################

Dear Reviewer, we added a description in limitations:

“In addition to that, it should be acknowledged that this study is a narrative review, not a systematic review. Therefore, the selection of studies is prone to bias and is based on authors’ judgments. The authors aimed to reduce the selection bias and described their selection criteria in Methods and Materials section. Moreover, the conclusion drawn in this review are susceptible to potential publication bias regarding cited literature [107]. To reduce it many databases were searched for relevant articles.

  1. Page 12 lines 433. It is not clear what exactly authors mean for ‘varied exploration establishes various successful frame works’.

######################

We wanted to emphasize that there are many successful methods for describing air pollution. However, for clarity, we deleted this sentence.

  1. Authors may want to provide information on which exposure measures have been applied (e.g., daily monitoring data; satellite-based modeling data) in the reviewed studies. This would help understand data availability and potential misclassification for exposure measurement of air pollution in the reviewed Polish studies.

######################

Dear Reviewer, we discussed that in the 6th point of your comments.

One more time, thank you for your valuable comments on our manuscript.

Reviewer 2 Report

Review

Air pollution in Poland. A 2022 update with focus on respiratory diseases

(ijerph-1540855 - 23.12.21)

The study is a very extensive review of scientific articles related to the level of atmospheric air pollution in Poland (mainly in urban agglomerations) and its impact on the health of residents. The authors are trying to formulate a general relationship binding the degree of air pollution with its impact on any type of respiratory disease. At the same time, they are trying to determine the directions of scientific research necessary to limit the high level of air pollution in Poland and its negative health effects.

Comments

  1. (L.38-40) According to the WHO report [7], the annual mean concentrations: 5 mg/m3 (PM2.5), 15 mg/m3 (PM10), and 10 mg/m3 (NO2) are the recommended air quality guidelines, that should be achieved in a long term period. But even the much less restrictive, the Interim 3 target values: 15 mg/m3 (PM2.5), 30 mg/m3 (PM10), and 20 mg/m3 (NO2), are also exceeded in most of the Polish agglomerations. On the other hand, to assess the seasonal air quality the short-term limit values should be rather applied, e.g. by the number of 24-h limit exceedances.

  1. (L. 75-82) It is not clear why in some papers the relative health impact of SO2 pollution is higher than this related to PM2.5, which is commonly considered to be the main source of health problems.

  1. A large number of acronyms used in the paper are not known to the average reader. For example, indices used to assess the population health burden (FVC, FEV1, MV, AP, IRV, COPP, etc.), as well as those related to parameters of statistical, should be defined and explained on a separate list (at the beginning or end of the study).

  1. (L. 347-367) The problem of correlation of air pollution level with COVID-19 cases, presented in Section 6.2, seems to be a more complicated one. In addition to air pollution, there are other factors whose impact on morbidity and mortality due to Covid-19 is probably greater than the air quality. These factors are e.g.:
  • definitely underestimated the official number of COVID-19 cases detected due to the very small (compared to the other EU countries) number of tests performed,
  • very low degree of vaccination of the country's population, resulting in a significant extent from the popularity of various “anti-vaccine theories”,
  • strong spatial diversity of vaccinated and not vaccinated populations.

     Therefore, the results on the correlation between COVID-19 and air pollution levels should be considered as qualitative, otherwise, much more complex analysis is required.

  1. With regard to public awareness (section 7), the numerous media relations show that in the case of the importance of vaccination to reduce the spread of the virus, general knowledge and public awareness are very low (obviously much lower than in the case of doctors), contrary to air pollution.

  1. The final Author’s conclusions about “the urgent need to reduce the air pollution level” as well as “an intensive promotion of awareness regarding this issue” are absolutely right, but an additional condition must be met in order for them to be implemented: The Polish government must actually start pursuing a pro-ecological policy and definitely change the fuel mix toward low-carbon solutions.

7. The reference list should be supplemented with the following two items

Katarzyna Maciejewska. Short-term impact of PM2.5, PM10, and PMc on mortality and morbidity in the agglomeration of Warsaw, Poland. Air Quality, Atmosphere & Health (2020). 13:659–672 https://doi.org/10.1007/s11869-020-00831-9

This paper considers the health effects of short-term episodes of high PM concentrations (PM2.5, PM10, PMc) in Warsaw. The population health endpoints encompass the daily counts of the hospital admissions and death cases (cardiovascular, respiratory, and all-cause). The statistical time series analysis is used for input data investigation. 

Łukasz Adamkiewicz, Katarzyna Maciejewska, Krzysztof Skotak, Michał Krzyżanowski,  Artur Badyda, Katarzyna Juda-Rezler and Piotr Dąbrowiecki. Health-Based Approach to Determine Alert and Information Thresholds for Particulate Matter Air Pollution. Sustainability (2021), 13, 1345. https://doi.org/10.3390/su13031345

The study presents health impact assessment methods as a tool to obtain the possible health benefits in Poland. The benefits result from reducing PM10 concentrations below a certain threshold level. Hospital admissions due to cardiovascular and respiratory diseases are considered as impacts of daily mean PM10 concentrations.

Author Response

Air pollution in Poland. A 2022 update with focus on respiratory diseases

(ijerph-1540855 - 23.12.21)

The study is a very extensive review of scientific articles related to the level of atmospheric air pollution in Poland (mainly in urban agglomerations) and its impact on the health of residents. The authors are trying to formulate a general relationship binding the degree of air pollution with its impact on any type of respiratory disease. At the same time, they are trying to determine the directions of scientific research necessary to limit the high level of air pollution in Poland and its negative health effects.

 ######################

Dear Reviewer, thank you for your valuable comments. They helped to improve our manuscript.

Comments

  1. (L.38-40) According to the WHO report [7], the annual mean concentrations: 5 mg/m3 (PM2.5), 15 mg/m3 (PM10), and 10 mg/m3 (NO2) are the recommended air quality guidelines, that should be achieved in a long term period. But even the much less restrictive, the Interim 3 target values: 15 mg/m3 (PM2.5), 30 mg/m3 (PM10), and 20 mg/m3 (NO2), are also exceeded in most of the Polish agglomerations. On the other hand, to assess the seasonal air quality the short-term limit values should be rather applied, e.g. by the number of 24-h limit exceedances.

 ######################

We added a description of interim targets and 24h limits:

In addition to that, even if the less restrictive “interim 3 target values are considered 15 μg/m3 (PM2.5), 30 μg/m3 (PM10), and 20 μg/m3 (NO2), the thresholds are still exceeded in the majority of polish cities [5,8]. Moreover, in 2021 the WHO recommends that the 24-hour mean daily concentrations should not exceed 15 μg/m3 (PM2.5), 45 μg/m3 (PM10), and 25 μg/m3 (NO2), respectively [8].

  1. (L. 75-82) It is not clear why in some papers the relative health impact of SO2 pollution is higher than this related to PM2.5, which is commonly considered to be the main source of health problems.

######################

We added the following sentences:

 “Interestingly, results for PM2.5 and PM10 were not significant [12], even though the change in PM2.5 concentration was shown to have the greatest strength of association with daily mortality rates [13,14]. However, rise of PM2.5 concentration by 10-µg/m3 was found to be a risk factor for increased cardiovascular deaths (RR 1.07; 95% CI 1.02-1.12) [12].”

  1. A large number of acronyms used in the paper are not known to the average reader. For example, indices used to assess the population health burden (FVC, FEV1, MV, AP, IRV, COPP, etc.), as well as those related to parameters of statistical, should be defined and explained on a separate list (at the beginning or end of the study).

######################

 We added a list at the end of the manuscript, after conclusions

  1. (L. 347-367) The problem of correlation of air pollution level with COVID-19 cases, presented in Section 6.2, seems to be a more complicated one. In addition to air pollution, there are other factors whose impact on morbidity and mortality due to Covid-19 is probably greater than the air quality. These factors are e.g.:
  • definitely underestimated the official number of COVID-19 cases detected due to the very small (compared to the other EU countries) number of tests performed,
  • very low degree of vaccination of the country's population, resulting in a significant extent from the popularity of various “anti-vaccine theories”,
  • strong spatial diversity of vaccinated and not vaccinated populations.

     Therefore, the results on the correlation between COVID-19 and air pollution levels should be considered as qualitative, otherwise, much more complex analysis is required.

 ######################

 We added a sentence in the section regarding COVID-19:

However, this results should be considered as qualitative. There is a number of factors, for example the spread of “anti-vaccine” theories, probable underestimation of the official number of COVID-19 cases as well as strong spatial diversity of vaccinated and not vaccinated populations that might have affected the exact estimation coefficients.

  1. With regard to public awareness (section 7), the numerous media relations show that in the case of the importance of vaccination to reduce the spread of the virus, general knowledge and public awareness are very low (obviously much lower than in the case of doctors), contrary to air pollution.

 ######################

Dear Reviewer, thank you for your comment. However, as the comparison of public awareness regarding the importance of vaccination and air pollution levels was not in the scope of our review, we did not change anything in the text. 

  1. The final Author’s conclusions about “the urgent need to reduce the air pollution level” as well as “an intensive promotion of awareness regarding this issue” are absolutely right, but an additional condition must be met in order for them to be implemented: The Polish government must actually start pursuing a pro-ecological policy and definitely change the fuel mix toward low-carbon solutions.

######################

 We added a sentence:

Therefore, in order to minimize the air pollution levels, a more advanced national pro-ecological policy must be implemented and a shift from coal towards low-carbon energy sources must be a endorsed.

  1. The reference list should be supplemented with the following two items

Katarzyna Maciejewska. Short-term impact of PM2.5, PM10, and PMc on mortality and morbidity in the agglomeration of Warsaw, Poland. Air Quality, Atmosphere & Health (2020). 13:659–672 https://doi.org/10.1007/s11869-020-00831-9

This paper considers the health effects of short-term episodes of high PM concentrations (PM2.5, PM10, PMc) in Warsaw. The population health endpoints encompass the daily counts of the hospital admissions and death cases (cardiovascular, respiratory, and all-cause). The statistical time series analysis is used for input data investigation. 

 ######################

 We cited the paper:

Maciejewska report that in Warsaw a short-term exposure to increased PM2.5 and PM10 concentrations by 10-µg/m3 results in increase of RR by 0.7% and 0.3%, respectively [11]

Łukasz Adamkiewicz, Katarzyna Maciejewska, Krzysztof Skotak, Michał Krzyżanowski,  Artur Badyda, Katarzyna Juda-Rezler and Piotr Dąbrowiecki. Health-Based Approach to Determine Alert and Information Thresholds for Particulate Matter Air PollutionSustainability (2021), 13, 1345. https://doi.org/10.3390/su13031345

The study presents health impact assessment methods as a tool to obtain the possible health benefits in Poland. The benefits result from reducing PM10 concentrations below a certain threshold level. Hospital admissions due to cardiovascular and respiratory diseases are considered as impacts of daily mean PM10 concentrations.

 ######################

We added a description of the study:

To increase the awareness of air pollution, alerts during days with relatively high pollution levels can be provided. Adamkiewicz et al. suggest to implement national information and alert thresholds equal to 64 µg/m3 and 83 µg/m3 of daily mean PM10 concentrations, respectively [75]. Theoretically, reduction of exposure to air pollution during this days would reduce the burden of hospital admissions attributable to air pollution by 75% and 50%, in regard to the best case scenario (no days with PM10 concentration exceeding 50 µg/m3).

Round 2

Reviewer 1 Report

Page 18 line 586. Please check the number of section for lung cancer. It seems to be 6.4 (or 6.5) not 6.

Author Response

Dear Reviewer,

we corrected the issue.

Thank you for your reviews.